# Increasing Energy and Material Consumption Efficiency by Application of Material and Energy Flow Cost Accounting System (Case Study: Turbine Blade Production)

**Asghar Hakimi** [1] ⬤ **, Zahra Abedi** [1],* **and Fatemeh Dadashian** [2]

[1] Department of Natural Resources and Environment, Science and Research Branch, Islamic Azad University, Tehran 1477893855, Iran; hakimi_asghar@yahoo.com

[2] Textile Engineering Department, Amirkabir University of Technology, Tehran 1591639675, Iran; dadashia@aut.ac.ir

* Correspondence: abedi2015@yahoo.com; Tel.: +98-91-2390-5928

**Abstract:** It is often difficult to extract data on material and energy wastes and related costs in the value chain of manufacturing products. Many organizations are not fully aware of the actual costs of material and energy wastes. For this purpose, advanced costing methods should be used. For this case study, we used material and energy flow cost accounting (MEFCA) to determine material costs, losses, and waste management in the manufacturing of turbine blades at the Iran Power Plant Company. Using the extracted data, the manufacturing costs of turbine blades were studied. The conventional method of turbine blades production is the machining method, which produces a significant amount of material and energy waste. By studying different methods, we found that there is an alternative method called forging, which reduces losses and costs. Finally, the costs of the two methods were compared. Engineering economics techniques were also used to compare the two methods on a long-term planning horizon.

**Keywords:** MEFCA; cost method; turbine blade manufacturing; environmental management accounting

## 1. Introduction

Today, energy consumption at the global level is increasing continuously, leading to substantial carbon dioxide emissions and serious environmental problems [1,2]. As a result, the importance of energy consumption management, especially in the manufacturing sector, is clear. Given the limited resources in the world, material consumption management will also be of particular importance.

Producers of goods and services are trying to control material and energy flows [3]. Material and energy flow cost accounting is regarded as one of the most powerful accounting tools for environmental management. This approach is an effective method used to address the need to increase productivity and reduce environmental impacts by promoting transparency in the use of materials and resources [4]. MEFCA is an accounting method that measures all material and energy flows in physical and monetary units. In addition, this approach includes costs associated with financial products and losses [5]. The application of this method is independent of the type of production system or organization. The only requirement is for the company to be a consumer of materials and energy in its processes [4]. The MEFCA method divides the entire production system into quantitative centers (QCs). QCs are part of a production system in which inputs and outputs must be physically determined and denominated in monetary units. These areas usually include locations that change or store materials [4,6]. The QC is the starting point for collecting data on physical units in terms of resource measurement. The material and energy used for each QC must be measured in physical units. Then, all QC information must be aggregated into a flow model.

In this research, implementation of the MEFCA system in Iran's power industry was proposed. Using the proposed method can improve the efficiency of material usage, reduce losses, and improve waste management in the power industry. The proposed method was implemented in the typical turbine blade manufacturing process of the country's power plant repair company.

The innovations of this study can be summarized as follows:

- Applying a new costing evaluation system in the power plant industry;
- Improving the efficiency of raw materials and energy usage by applying the MEFCA method in one of the country's power plant repair companies;
- Saving energy and reducing costs by avoiding material losses and hidden costs of waste;
- Using engineering economics techniques and cash flow present value method in a long-term planning horizon.

The structure of this article includes the following sections: First, the research background is reviewed and the importance and necessity of research are explained. Then, the proposed methodology for improving the efficiency of materials and energy is presented. In the following section, the results of the simulation and case study are analyzed; finally, the conclusions regarding the proposed method are presented.

## 2. Literature Review

Energy is one of the most expensive inputs available to industries worldwide. However, the price of energy in Iran was relatively low until 2012 due to the payment of energy subsidies by the government. Now, given the Iranian government's targeted subsidies and price liberalization, energy prices are expected to rise sharply. This increase in price will increase production costs in industries, especially energy-intensive industries such as steel, automotive, cement, petrochemical, etc. In proportion to the share of energy is the cost price of goods, wherein the price of goods produced increases [7]. In a previous study, technical efficiency and energy efficiency (as a special input) of the country's petrochemical industry during the years 1994–2008 was evaluated and analyzed using the data envelopment analysis method [8].

The material and energy flow cost accounting (MEFCA) method is considered one of the most powerful costing tools for environmental management. This method is an effective approach to addressing the need to increase productivity and reduce environmental impact by promoting transparency in the use of materials and resources [9]. The previous version of the MEFCA method was the material flow cost accounting (MFCA) method, which has been described as among the most basic of the environmental management accounting (EMA) tools. The data afforded by MFCA also provide a foundation for the development of further environmental management accounting activities, which may include investment appraisal, environmental impact assessment, and short- and long-term environmental budgeting [10]. As energy efficiency is becoming a challenging task for researchers, organizations, and energy consumers [11], there was a need for a new approach that also includes energy costs. Primary energy consumption has accounted for more than 60% of global $CO_2$ emissions, leading to an increase in global warming [12], highlighting the importance of this subject.

This new method is known as MEFCA and has been proven to have many applications in various industries [13]. Behnami et al. used this method in a petrochemical wastewater treatment plant [13]. In their research using this novel stepwise approach, decision-makers were more confidently able to enhance both financial and environmental performance and to define appropriate improvement plans. The usage of this method in variant productions with large-scale plant manufacturers was discussed by Schmidt et al. [14]. In Germany, the MEFCA was developed many years ago to cope with the limitations on resources needed. MEFCA was first used in Japan on a large scale. ISO standards have already provided methods for this purpose [15]. According to Chinese companies, standard costs are used to calculate production costs for decision-making and as a contribution to budgeting. Their standards are based on the efficiency of past experiences and their annual review of

standards [16]. Material flow cost accounting can be discussed as a potential approach to illustrate the quantitative and monetary impacts of material flow management [17].

MEFCA was applied at the largest ceramic tile factory in the Czech Republic, Lasselsberger. Research demonstrated the importance of data obtained from the MEFCA system and their application to optimize production processes for the specific requirements of a company's production process [18]. Using the MEFCA concept, a simulation model for the supply chain, including a Japanese gear manufacturer and its customers, was developed to visualize the large amount of waste generated by the manufacturing process, providing environmental and economic benefits to the entire supply chain [19].

The material and energy flow cost accounting system is one of the tools that is essential to the three principles of reduction, reuse, and recycling to increase productivity, reduce costs, and sustain production and consumption. This system was also implemented in a factory unit in Azerbaijan. After the implementation of this system, the highest and lowest wastage of materials and energy were estimated in treatment and packaging centers, respectively. Solutions were then introduced to reduce production costs, waste, and environmental damage. The effectiveness of the MEFCA approach was investigated in a paper production company in KwaZulu-Natal, South Africa, which, based on the evidence and results of the managers; the researchers concluded that the company should integrate the MEFCA method with the current system to ensure that it is sustainable in the future [20].

The reasons why manufacturing companies are looking for sustainable resources include lack of resources, environmental awareness, and the potential for cost savings. To address these issues, MEFCA has been employed as an environmental management tool. According to the MEFCA, a printed circuit board (PCB) production study systematically generated several linear cost calculation models during the production process to capture actual effluent flows as well as a cost-benefit analysis, demonstrating that this research could improve production. Making the system profitable and sustainable by improving resource productivity can also provide decision support for PCB production [21].

Small- and medium-sized companies around the world are often challenged by issues related to increasing material and energy efficiency, waste management, and sustainability. The MEFCA method is used as a tool for optimal productivity to overcome these challenges [22]. The MEFCA method has also been used in the food supply chain; with the help of this method, the potential for cost reduction, revenue generation, and reduction in carbon dioxide due to the prevention and disposal of food waste in the production of blackberry juice in Germany was obtained [23].

This literature review shows that the MFCA technique and its more advanced version, the MEFCA method, are both valuable for the environmental assessment of production processes. A review of past research and previous experiences in the field of turbine blade production and its costing revealed that the method used to produce turbine blades has not been evaluated using the MEFCA method until this study. The MEFCA method has been used in many studies to estimate material and energy waste costs. In this research, in addition to this issue, it was used as a decision-making method for comparison between different manufacturing methods. The study of different methods for manufacturing turbine blades has also been considered by researchers. For example, Echin and Bondarenko [24] studied the technical specifications of manufacturing turbine blades using the forging method. Torres et al. [25] also studied the method of manufacturing turbine blades using laser melting. In another study, Cygan [26] technically analyzed the manufacturing of a turbine blade using 3D printing. However, the evaluations in these studies were mostly technical and did not address the issue of energy and material costs and wastages. Only Torres et al. [25] conducted environmental analysis based on the carbon emission index.

## 3. Methodology

As stated in previous sections, the method used for this study was material and energy flow cost accounting. The structure of this method is shown in Figure 1 and consists of a main input, positive output, and output losses.

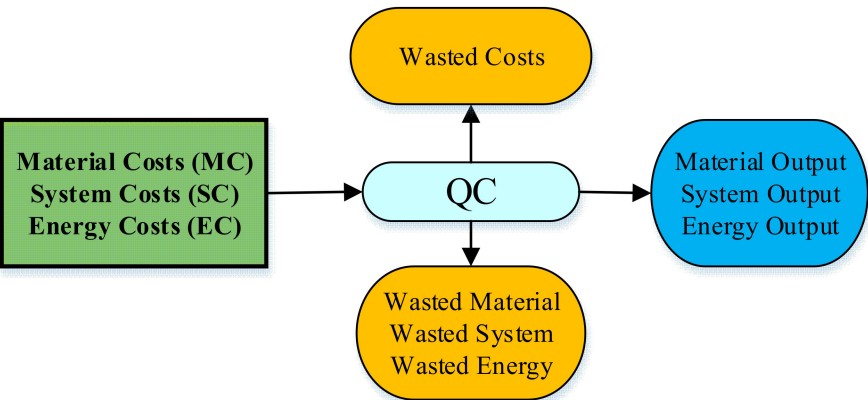

**Figure 1.** Material and energy flow cost accounting (MEFCA) structure.

The inputs of the system include raw materials; energy, which includes electricity and water; and the system, which includes tools and staff costs.

Losses include of raw materials, energy, and system losses.

Waste management costs: Each step includes costs for waste and sewage disposal.

Cost of positive output product: The final product of the raw material included the energy and system cost and the remaining positive raw material.

In this method, we divide the production process into several steps, each called a quantitative center. The symbols used here are listed in the Table 1. In this table, indexes for material costs (MC), system costs (SC), energy costs (EC), material weight (W), wasted material costs (WM), wasted system costs (WS), wasted energy costs (WE), positive material output costs (PM), positive system output costs (PS), and positive energy output costs (PM) are provided.

**Table 1.** Symbols used in this paper.

| Index | Symbol |
|---|---|
| input materials, system, and energy index | $index_{MC}$, $Index_{SC}$, $Index_{EC}$ |
| input materials, system, and energy cost | $Cost_{MC}$, $Cost_{SC}$, $Cost_{EC}$ |
| material weight and material waste | $W_{MC}$, $W_W$ |
| materials, system, and energy waste cost | $Cost_{WM}$, $Cost_{WS}$, $Cost_{WE}$ |
| materials, system, and energy positive output cost | $Cost_{PM}$, $Cost_{PS}$, $Cost_{PE}$ |

For each step, we allocate a coefficient for the quantitative center.

$$index_{MC} = \frac{cost_{MC}}{w_{MC}} \tag{1}$$

$$index_{SC} = \frac{cost_{SC}}{w_{MC}} \tag{2}$$

$$index_{EC} = \frac{cost_{EC}}{w_{MC}} \tag{3}$$

where MC indicates material cost, SC is system cost, and EC is energy cost. The wastage is calculated as follows:

$$cost_{WM} = w_W \times index_{MC} \tag{4}$$

$$cost_{WS} = w_W \times index_{SC} \tag{5}$$

$$cost_{WE} = w_W \times index_{EC} \tag{6}$$

The cost of the positive output is calculated as follows:

$$cost_{PM} = (w_{MC} - w_W) \times (index_{MC}) \tag{7}$$

$$\text{cost}_{PS} = (w_{MC} - w_W) \times (index_{SC}) \tag{8}$$

$$\text{cost}_{PE} = (w_{MC} - w_W) \times (index_{EC}) \tag{9}$$

The case study in this research was a power plant service company. In this study, using the MEFCA method, we intended to manage the cost of raw materials and energy in producing a complete set of turbine blades. In order to produce turbine blades, a series of complicated tasks should be undertaken. For this, raw material, alloy steel, enters the production line in large cubes. To manufacture the blades, these cubes must be cut to a certain size. These raw materials are then tested using non-destructive tests. The test cubes enter the CNC machining step to form the desired blades; the highest losses are incurred at this step. At the polishing step, products are polished with a blade grinding machine and then passed through to the quality control step. The QCs needed to produce a set of turbine blades are shown in Figure 2.

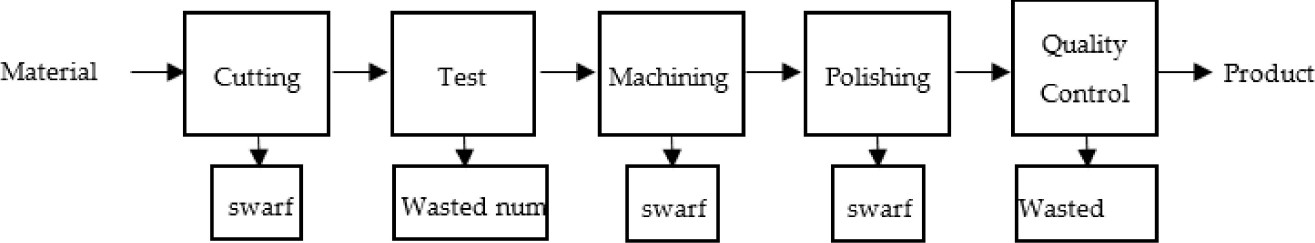

**Figure 2.** MEFCA structure for turbine blade production with machining method.

We used net present value (NPV) for the economic analysis. The NPV of the investment is the difference between the present value of the benefits and the costs resulting from an investment. NPV is a strong criterion used to determine if a project is profitable or not. It considers the interest rate (r), which is usually equal to the inflation rate; therefore, the real value of money at every year of operation is considered. NPV basically measures the dollar benefit (added value) of the project to the shareholders but it does not provide information on the safety margin or the amount of capital at risk [27]. For example, if the NPV of a project is calculated to be USD 2 million, it does not indicate the safety margin of the project. In contrast, investment or internal rate of return (IRR) measures the annual rate of return and provides safety margin information. Overall, for mutually exclusive projects and ranking purposes, NPV is always superior to IRR. Unfortunately, in the oil and gas industry, IRR is quite often used for making critical decisions. It is recommended to calculate and understand IRR methodology for each project. However, the ultimate decision whether to perform a project should be determined using NPV calculation [28].

## 4. Results

Table 2 shows the costs of producing a set of turbine blades by the machining method. The amount of raw materials is 3993.75 kg, which costs 399.3 units (in this research, each unit costs USD 400). The output is 518.57 kg, indicating a loss of 3475.17 kg, which is a significant amount. Given this amount of wastage, energy and system losses will also be large. In this article, a conversion rate of IRR 250,000 to USD 1 was used. The details of this method are given in Appendix A.

**Table 2.** Cost of turbine blade production with machining method.

| Process Step | Cost MC | Cost Energy | Cost SC | Negative MC Cost | Waste EC Cost | Waste SC Cost | Positive MC Cost | Positive EC Cost | Positive SC Cost | Waste Management |
|---|---|---|---|---|---|---|---|---|---|---|
| Cutting | 399.38 | 9.1 | 59.06 | 3.99 | 0.09 | 0.59 | 395.38 | 9.01 | 58.47 | 0.1 |
| Test | 395.38 | 3.12 | 36.17 | 2.58 | 0.02 | 0.24 | 392.81 | 3.1 | 35.93 | 0.1 |
| Machining | 392.81 | 19.74 | 393.47 | 340.01 | 17.09 | 340.58 | 52.79 | 2.65 | 52.88 | 0.5 |
| Polishing | 52.79 | 4.91 | 31.86 | 2.64 | 0.02 | 0.16 | 52.53 | 4.89 | 31.7 | 0.15 |
| Quality Control | 52.53 | 3.12 | 91.44 | 0.67 | 0.04 | 1.17 | 51.86 | 3.08 | 90.27 | 0.1 |
| Total | 399.38 | 40 | 612 | 347.52 | 17.26 | 342.74 | 51.86 | 22.74 | 269.26 | 0.95 |

The machining step has the highest amount of wastage, which is approximately 86.5% of the total input material.

One of the alternative methods of producing blades is the forging method. In this method, the cubes are cut to the desired blade size, then pressed into a premade mold using a press machine, and continuing the same steps as in the machining method. Figure 3 illustrates the steps of the blade production process.

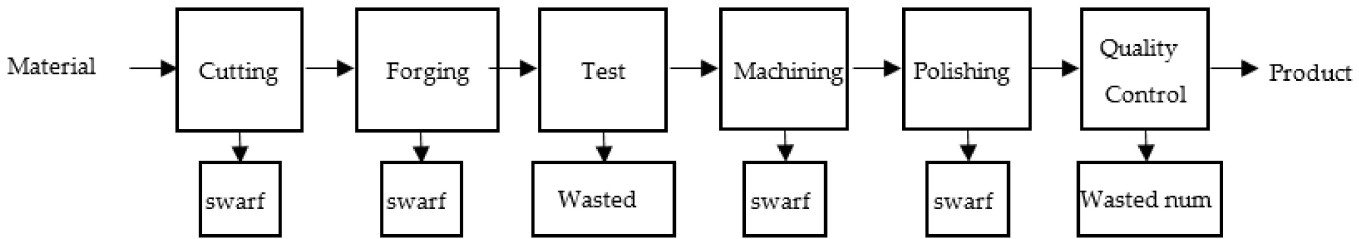

**Figure 3.** MEFCA structure for turbine blade production with the forging method.

The costs of this method are shown in Table 3. According to the data in this table, the amount of inputs is 1312.5 kg, of which 525.02 kg is positive output. As a result, its wastage is 787.47 kg, which is little compared with the waste produced using the machining method. The details of this method are provided in Appendix B.

**Table 3.** Cost of turbine blade production with the forging method.

| Process Step | Cost MC | Cost Energy | Cost SC | Negative MC Cost | Waste EC Cost | Waste SC Cost | Positive MC Cost | Positive EC Cost | Positive SC Cost | Waste Management |
|---|---|---|---|---|---|---|---|---|---|---|
| Cutting | 131.25 | 5.30 | 34.36 | 1.31 | 0.05 | 0.34 | 129.94 | 5.24 | 34.01 | 0.07 |
| Forging | 129.94 | 7.28 | 120.00 | 12.99 | 0.73 | 12.00 | 116.94 | 6.56 | 108.00 | 0.30 |
| Test | 116.94 | 2.08 | 23.36 | 2.27 | 0.04 | 0.45 | 114.68 | 2.04 | 22.91 | 0.10 |
| Machining | 114.68 | 10.23 | 226.53 | 61.88 | 5.52 | 122.23 | 52.80 | 4.71 | 104.29 | 0.20 |
| Polishing | 52.80 | 4.86 | 31.53 | 0.11 | 0.01 | 0.06 | 52.69 | 4.85 | 31.47 | 0.10 |
| Quality Control | 52.69 | 3.12 | 91.44 | 0.19 | 0.01 | 0.33 | 52.50 | 3.11 | 91.11 | 0.10 |
| Total | 131.25 | 32.87 | 527.22 | 78.75 | 6.36 | 135.42 | 52.50 | 26.51 | 391.79 | 0.87 |

Now, we compare the two methods used to manufacture turbine blades. According to Table 4, the input cost of machining is 1051.36 units, while for the forging method, 691.33 units, which is reduction in input cost of 360.03 units and in the material amount to 2681 kg. The cost of machining is USD 708, while in the forging method it is USD 220, which is a decreased of 488 units. Moreover, the cost of positive output is USD 343, which covers

32% of the total cost of production and nearly 68% is lost, while in the forging method, 68% of the input cost is positive and 32% is wastage. The cost of waste management also decreases with the forging method.

**Table 4.** Comparison of two methods of turbine blade production.

| Process Method | Waste | | Positive Output | | | | Negative Output | | | | Input | | | |
|---|---|---|---|---|---|---|---|---|---|---|---|---|---|---|
| | WM | MC | MC (kg) | EC | SC | MC | EC | SC | MC (kg) | MC | EC | SC | MC (kg) |
| Machining | 0.95 | 51.857 | 518.572 | 22.735 | 269.257 | 347.517 | 17.261 | 342.739 | 3475.178 | 399.375 | 39.996 | 611.997 | 3993.75 |
| Forging | 0.87 | 52.502 | 525.025 | 26.507 | 391.793 | 78.747 | 6.361 | 135.422 | 787.47 | 131.25 | 32.86 | 527.216 | 1312.5 |
| Improvement percentage | 8% | 1% | 1% | 17% | 46% | 77% | 63% | 60% | 77% | 67% | 18% | 14% | 67% |

To illustrate the comparison between the machining and forging methods, Tables 2–4 and Figures 4–6 are provided below.

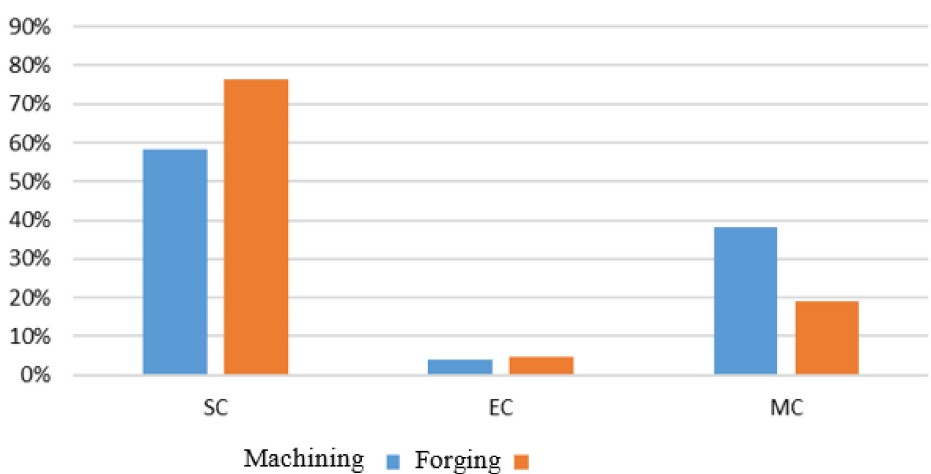

**Figure 4.** Comparison of the total percentage of inputs for the two methods.

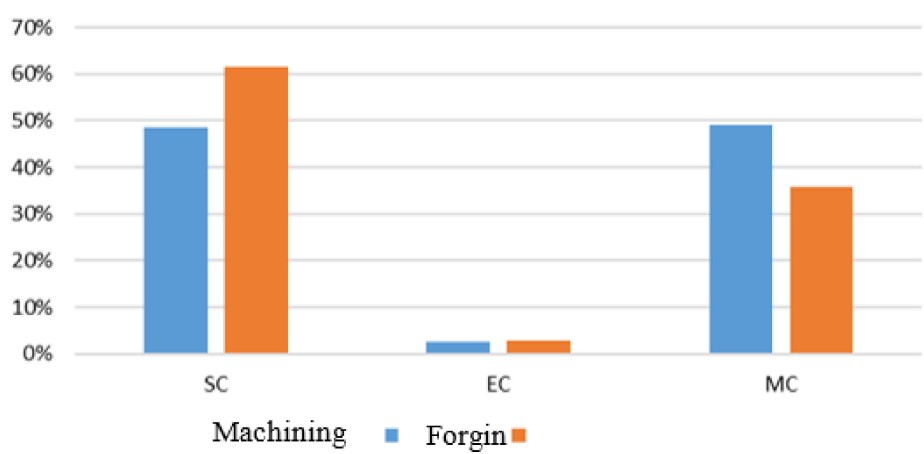

**Figure 5.** Comparison of the total percentage of negative output for the two methods.

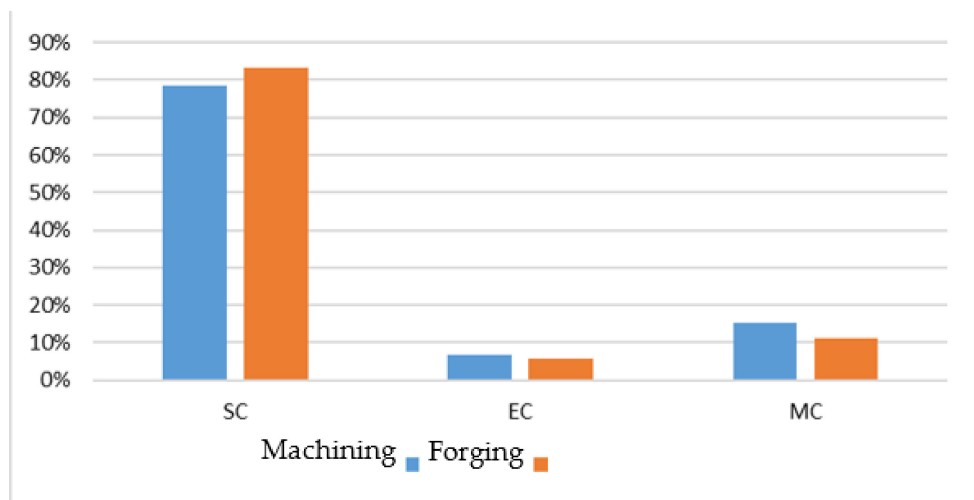

**Figure 6.** Comparison of the total percentage of positive output for the two methods.

As shown in the above diagrams, the proportion of system costs in the forging method is higher than machining, but the energy costs are about the same. For material costs, the costs are higher in the forging method.

By comparing the percentage of total negative outputs between the two methods, it is evident that the system costs are higher in the forging method, but the material costs in the machining method are lower.

In addition, we investigated the economic cost of turbine blade production using engineering economics techniques. The financial information used and past data for these calculations are shown in Table 5.

**Table 5.** Financial information needed to create financial flows [7,25].

| No. | Item | Amount |
|---|---|---|
| 1 | Production capacity of blades in one year | 15 sets |
| 2 | Average selling price | USD 0.6 million |
| 3 | Present value of machining equipment | USD 60 million |
| 4 | Service remaining life of machining equipment | 40 years |
| 5 | Sacrificial value of machining equipment | USD 4 million |
| 6 | The amount of capital needed to start a forging workshop | USD 4 million |
| 7 | Service life of forging equipment | 40 years |
| 8 | Sacrificial value of forging equipment | USD 0.2 million |

The following steps are used to compare the two methods of forging and machining in engineering economics techniques:

- Planning horizons (here, 40 years with four-year time periods, for a total of 10);
- Determination of financial flows related to initial investment (II);
- Determination of the financial flows related to the equipment's sacrificial value (SV);
- Determination of the cash flow of incomes (I);
- Determination of operating cost cash flow (OC);
- Calculation of cash flow before tax (CFBT);
- Depreciation calculation (D);
- Calculation of taxable income (TI);
- Tax calculation (T);
- Calculation of cash flow after tax (CFAT);
- Calculation of net present value of cash flow (NPV).

The initial investment in the first planning period was USD 60 million, while the cost of equipment after the end of life was USD 0.2 million. Given that each period was four years and the production capacity was 15 sets of blades per year, the number of blades produced was 60 sets over a period, which will generate an income of USD 36 million. Given the current inflationary situation, this revenue increases by 50% for each four-year period. The same is true of operating costs. Operating costs, including total energy, system, and material costs for machining according to the calculations in the previous research, amounted to USD 0.42 million for 1180 blades, which would be USD 21.375 million for production of 60 sets of blades. In the forging method, similarly, operating costs were USD 14.05 million.

Depreciation by the straight-line method for each four-year period equaled USD 5.6 million for machining equipment and USD 14.95 million for the forging method. Tax was calculated on the basis of the tax rate of 9%. The amount of taxable income was calculated followed by the amount of tax. The cash flow after tax was then calculated. Finally, the present value of cash flow was determined and compared between the two methods. The interest rate was 20%. Higher present value in this comparison means that the method is better in terms of economic justification.

Using these calculations, the net present value of the machining method was USD 0.31 billion and USD 0.484 billion for the forging method. These numbers represent a significant difference between these two approaches in terms of engineering economics techniques. As a result, it can be argued that the forging method is far better in terms of a long-term planning horizon.

## 5. Conclusions

Material and energy flow cost accounting (MEFCA) is the organizational tool used by manufacturing companies to improve the efficiency of their raw materials, energy, and systems. MEFCA's goal is to save energy and money by avoiding wastages. To achieve this, the MEFCA can be used to calculate the actual cost of waste, including hidden costs. MEFCA is an important element of operating resource productivity for companies and is standardized through ISO 14051.

There are two different methods to produce turbine blades that were technically analyzed in this case study. One of these methods is forging, which can be used instead of the conventional method of machining as it is technically and economically justified based on the findings of this study. However, it also had to be considered from an environmental perspective. We performed this important step using the MEFCA technique.

We used the MEFCA method to study and compare the two methods of blade production. The input for the machining method was 3993.5 kg, which resulted in 87% material wastages. Using the MEFCA algorithm, we determined that most losses were related to the CNC machining step. The solution to this problem was to break the pieces down to 1312.5 kg of raw material and to reduce the loss to 60% of the input (787.5 kg). In the present study, using deduced data, we investigated the costs of turbine blade production. The conventional method of producing a turbine blade is the machining method, which results in significant losses. By studying the different methods, we found that the forging method reduces losses and costs. In the end, we compared the costs of the two methods. Engineering economics techniques were also used to compare the two methods over a long-term planning horizon. The MEFCA method can be used in all industries to cost a product, which, in the future, will result in lower losses and cost savings.

**Author Contributions:** Conceptualization, A.H. and Z.A.; methodology, A.H.; software, A.H.; validation, Z.A. and F.D.; formal analysis, A.H.; investigation, Z.A.; writing—review and editing, F.D.; visualization, A.H.; supervision, Z.A.; project administration, Z.A.; All authors have read and agreed to the published version of the manuscript.

**Funding:** This research received no external funding.

**Institutional Review Board Statement:** Not applicable.

**Informed Consent Statement:** Not applicable.

**Data Availability Statement:** Not applicable.

**Conflicts of Interest:** The authors declare no conflict of interest.

## Appendix A

In the machining method, there are five steps (cutting, testing, machining, polishing, and quality control); we provide details in the following figures.

Step 1

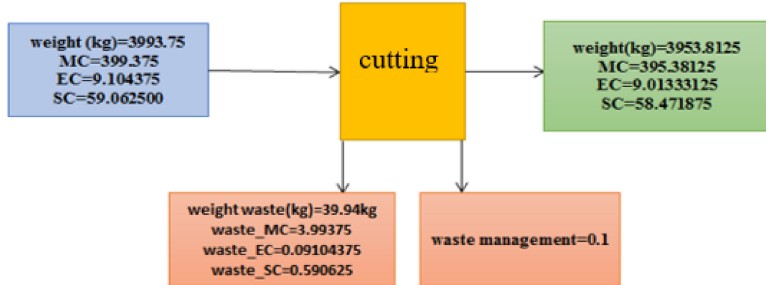

**Figure A1.** Cutting unit costs in the machining method.

Step 2

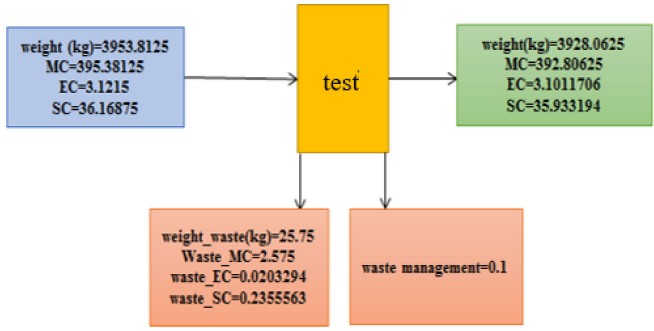

**Figure A2.** Test unit costs in the machining method.

Step 3

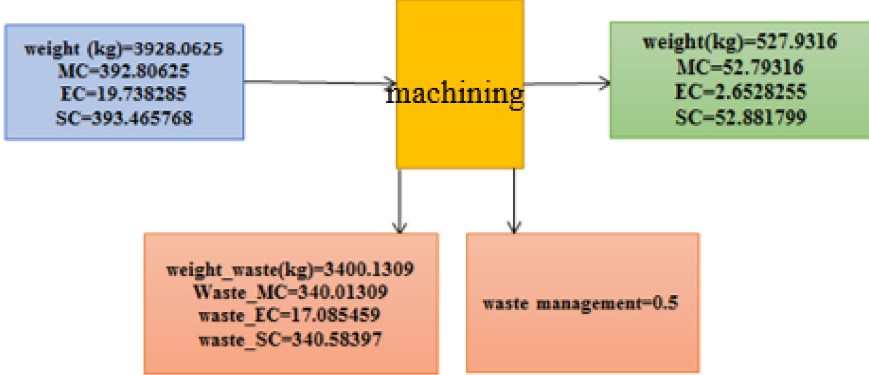

**Figure A3.** Machining unit costs in the machining method.

Step 4

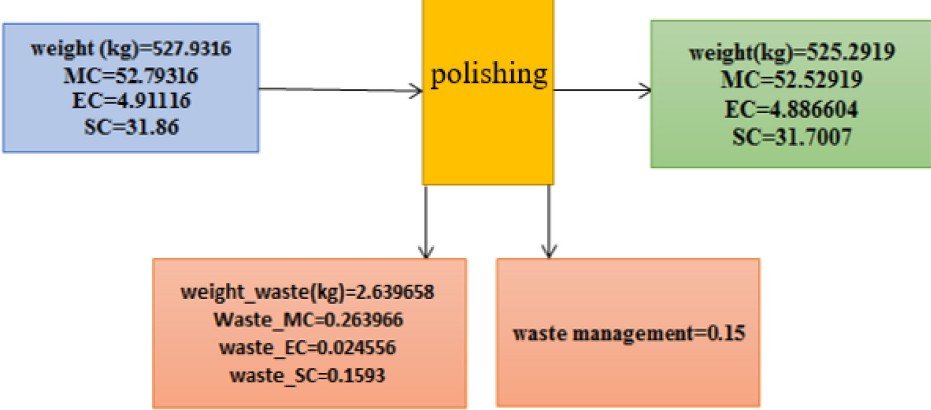

**Figure A4.** Polishing unit costs in the machining method.

Step 5

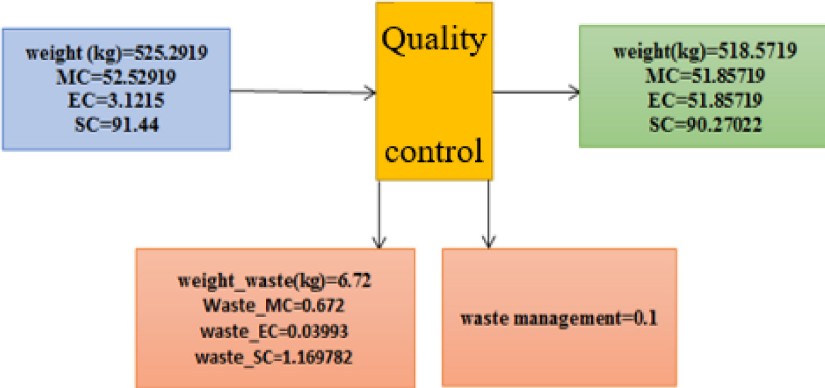

**Figure A5.** Quality control unit costs in the machining method.

As shown in the above results, there is a balance between inputs and outputs.

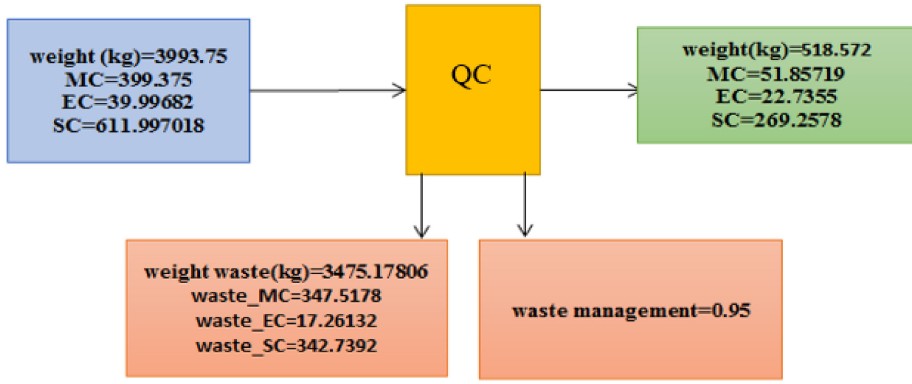

**Figure A6.** The sum of the costs of the various steps in the machining method.

**Appendix B**

There are six steps (cutting, forging, testing, machining, polishing, and quality control) in the forging method and some details are provided in the following figures.

Step 1

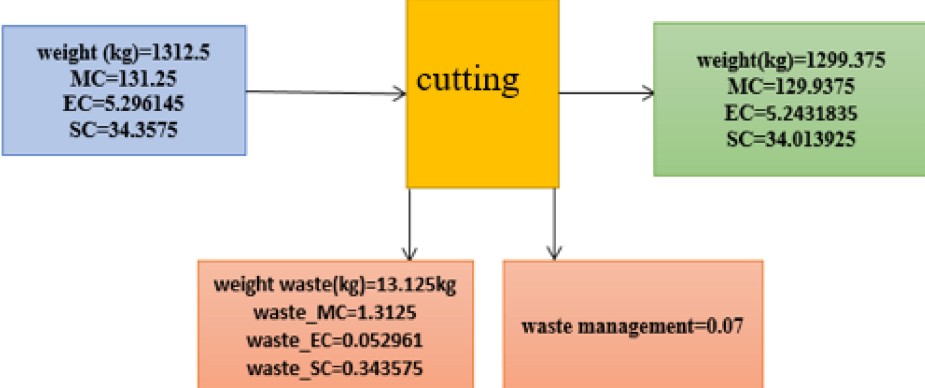

**Figure A7.** Cutting unit costs in the forging method.

Step 2

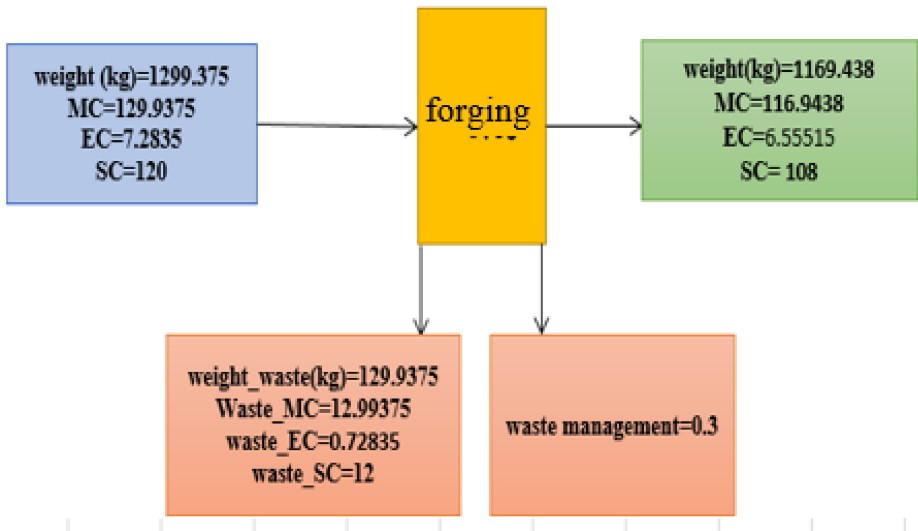

**Figure A8.** Forging unit costs in the forging method.

Step 3

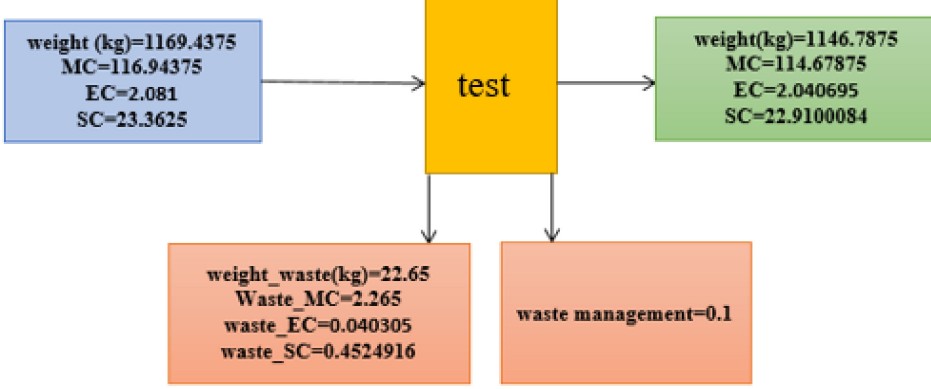

**Figure A9.** Test unit costs in the forging method.

Step 4

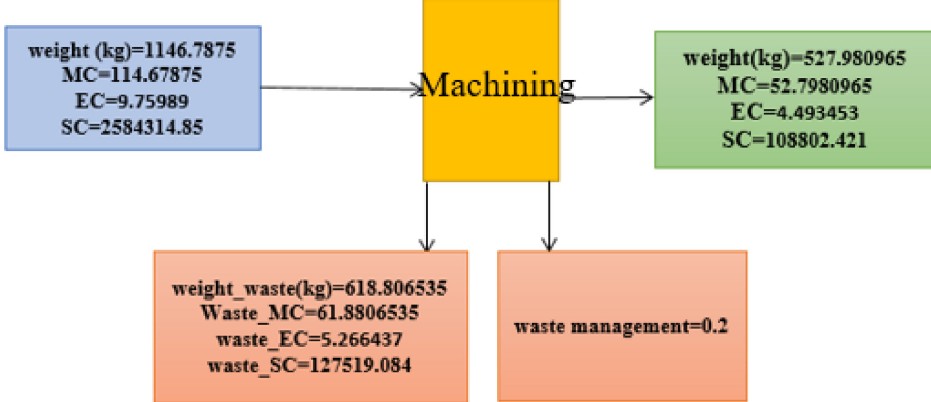

**Figure A10.** Machining unit costs in the forging method.

Step 5

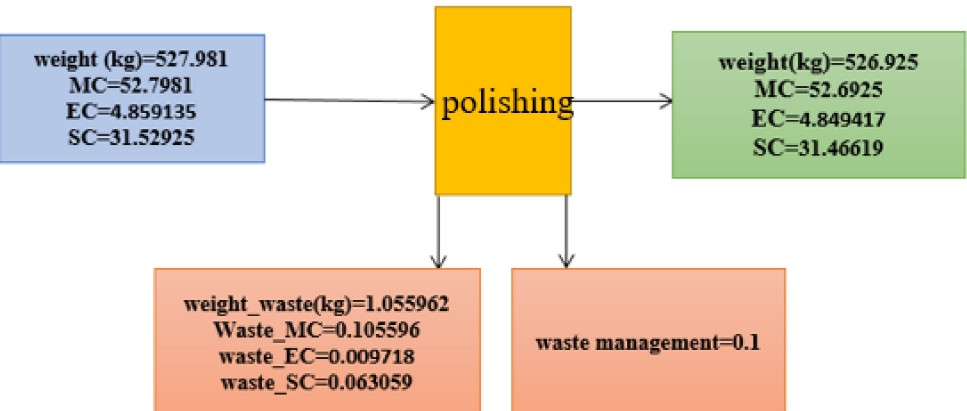

**Figure A11.** Polishing unit costs in the forging method.

Step 6

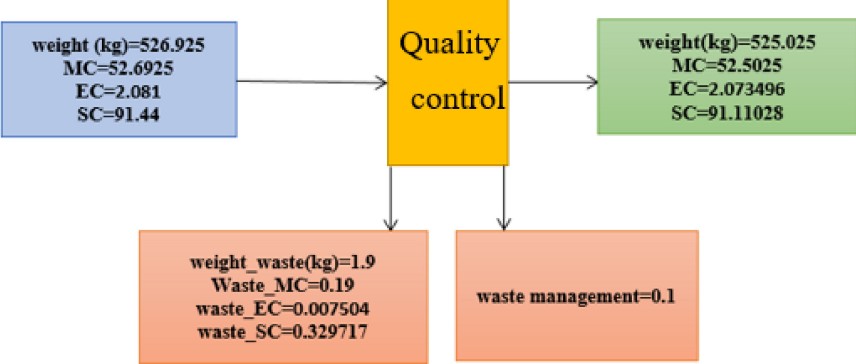

**Figure A12.** Quality control costs in the forging method.

In the forging method, there is a balance between input and output values.

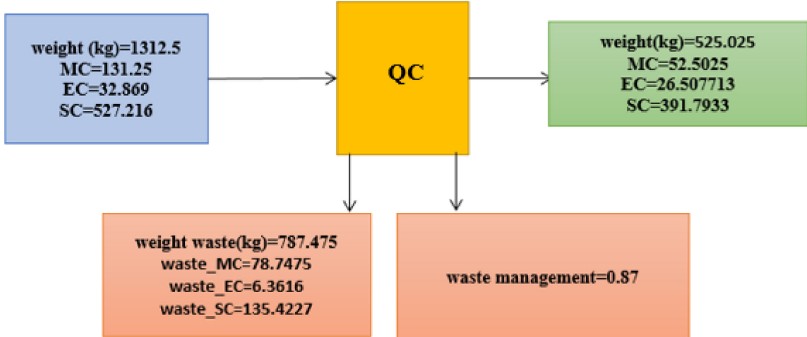

**Figure A13.** The sum of the costs of the various steps in the forging method.

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
