# Peer review of "Increasing Energy and Material Consumption Efficiency by Application of Material and Energy Flow Cost Accounting System (Case Study: Turbine Blade Production)"

_sustainability, doi:10.3390/su13094832_

Round 1
Reviewer 1 Report
Authors studied the manufacturing costs of turbine blades using the extracted data. They studied existing methods and found an alternative method called forging, which reduces losses and costs.
I found the following formal imperfections:
- figure 2. missing text "materi" and "produ"
- figure 3. missing text "produc"
- table 4 - improvement percentage - delete line between this two words
- recent literature will be better for this study
- figure 4., 5. and 6. - larger font of words under the graph
- please check all abbreviations
Author Response
Authors sincerely appreciate the careful attention to the present work. Comments that we received are really precise and explicit and addressing the thorough insight of reviewers to the intended subject. Suggestions and mentions of reviewers really helped us improve the manuscript and provide a paper of higher quality suitable for the publication in this respected journal. All changes and modifications are highlighted in the new version of the manuscript. All comments have been attended and answered. Responses are provided in this document.

Reviewer 2 Report
Here are some considerations:
1) In the introduction, the aim should be emphasized and justified. I have not found a clearly defined purpose of the article and research.
2) The research background section should be replaced, for example, with "literature review - research background". In this section, you should make a critical analysis of the world literature. On its basis you should show the so-called research gaps and justify why research and its analysis can fill these research gaps.
3) The methodology should include a detailed description of the data source. There is no research hypothesis (or thesis) that will be verified by the research and its analysis. Research questions can be used instead of a hypothesis.
4) A "discussion" section should be added and it should refer to the level of verification of the previously formulated hypothesis (or thesis) or research questions.
Author Response

(The authors gave the same response as above.)

Reviewer 3 Report
It is recommended to publish in the form as it was prepared.
Author Response

(The authors gave the same response as above.)

Reviewer 4 Report
Thanks for your manuscript and congrats for your work.
Please let me sugest some changes that, in my opinion, could improve the text.
In the first line of introduction refer the IEA document.
Line 10 of Research Background: please exchange the text "This method is known as MEFCA. This method has shown..." to (or similar) " This method is known as MEFCA and has shown..."
In figure 1, the material output from the quantitative center is the same as material with is also an output. One of them is losses. Could it be material output losses?
Material costs (MC) appears two times below figure 1.
Also in figure 1, there is only waste cost, but in the table there are material, energy and system waste.
In table there are no losses?
Please, redo fig 1 and better explain outputs.
lines 4 and 5 of table 1 are outputs. Please remove the input word from the index.
Block names in figures 2 and 3 are cut.
The cost units are in Rials. Is it possible to also express them in Euro or $ to allow future comparisons possible.
There is a negative cost in table 2 which is not explained.
Table 2 is difficult to understand.
Tables and figures are not referenced in the text. When you write "following table" or "next figure" please explicit the table or figure number.
In the economics comparison, there are no references to equipment costs or lifetime, no reference to the interest rate, ...
Introduction and research backgroud are correctly written and explained. However, metodology, results and conclusions still need a lot of work.
Please continue the good work.
Author Response

(The authors gave the same response as above.)

Round 2
Reviewer 2 Report
The authors took into account all my comments. I have no more comments.Author Response
First of all, we would like to appreciate the respectable reviewer for his/her constructive comments that enhanced the paper. We hope that our point-by-point answers address the reviewer’s concerns. All the changes in the revised paper are highlighted in blue.

Reviewer 4 Report
Thank you for all your work and acceptance of our comments.
The changes made in the manuscript are valuable and trully improve the reading and understanding of the text.
Let me write some comments which, for me, could improve the article.
Methodology is now consistent and correctly explained. However, results are too straightforward and Table 2 should add more information. Before constructing table 2, i would like to see the materials, energy and system on their units (mass, energy and monetary cost). By showing just the table after the MEFCA process, a lot of insights are lost and almost impossible to replicate. Is it possible to include a table before table 2 that explains inputs and outputs on their units?
Also, on table 2, the column Positive MC cost shoud be the first on the positive costs (materials, energy and system). The same on tables 3 and 4. Please follow a pattern.
A table for mass, energy and system for forging will also be importante to include before results.
There is no discussion on blade quality between forging and machining.
The economics comparison is, again, too superficial and lack comments, references on the written values.
Continue the good work.
Author Response
First of all, we would like to appreciate the respectable reviewer for his/her constructive comments that enhanced the paper. We hope that our point-by-point answers address the reviewer’s concerns. All the changes in the revised paper are highlighted in blue.
